# Rapid access to discrete and monodisperse block co-oligomers from sugar and terpenoid toward ultrasmall periodic nanostructures

Takuya Isono [1✉], Ryoya Komaki[2], Chaehun Lee[2], Nao Kawakami[2], Brian J. Ree[2], Kodai Watanabe[2], Kohei Yoshida[2], Hiroaki Mamiya[3], Takuya Yamamoto [1], Redouane Borsali[4], Kenji Tajima [1] & Toshifumi Satoh [1✉]

Discrete block co-oligomers (BCOs) are gaining considerable attention due to their potential to form highly ordered ultrasmall nanostructures suitable for lithographic templates. However, laborious synthetic routes present a major hurdle to the practical application. Herein, we report a readily available discrete BCO system that is capable of forming various self-assembled nanostructures with ultrasmall periodicity. Click coupling of propargyl-functionalized sugars (containing 1–7 glucose units) and azido-functionalized terpenoids (containing 3, 4, and 9 isoprene units) afforded the discrete and monodisperse BCOs with a desired total degree of polymerization and block ratio. These BCOs microphase separated into lamellar, gyroid, and cylindrical morphologies with the domain spacing ($d$) of 4.2–7.5 nm. Considering easy synthesis and rich phase behavior, presented BCO systems could be highly promising for application to diverse ~4-nm nanofabrications.

[1] Faculty of Engineering, Hokkaido University, Kita 13, Nishi 8, Kita-ku, Sapporo, Hokkaido 060-8628, Japan. [2] Graduate School of Chemical Sciences and Engineering, Hokkaido University, Kita 13, Nishi 8, Kita-ku, Sapporo, Hokkaido 060-8628, Japan. [3] National Institute for Materials Science, 1 Chome-2-1 Sengen, Tsukuba, Ibaraki 305-0047, Japan. [4] CERMAV, CNRS, Univ Grenoble Alpes, 38000 Grenoble, France. ✉email: isono.t@eng.hokudai.ac.jp; satoh@eng.hokudai.ac.jp

Ordered nanostructures such as lamellar (LAM), gyroid (GYR), cylindrical, and spherical morphologies generated by block copolymer (BCP) microphase separation are highly promising nanotemplates for the low-cost, large-scale, and high-throughput production of sub-10 nm nanomaterials and nanodevices[1–4]. In particular, directed self-assembly (DSA)-assisted BCP lithography has attracted considerable attention as a next-generation nanopatterning technology because it is more cost-effective and scalable than competing approaches[5–8]. One of the key challenges in this field is the development of BCP systems for the formation of highly ordered ultrasmall microphase-separated structures, enabling nanopatterning of a few nanometers.

Previous studies have shown that decreasing the degree of polymerization ($N$) while increasing the Flory–Huggins interaction parameter ($\chi$) is the most feasible and reliable method for achieving periodic microphase-separated structures with a domain spacing ($d$) <10 nm[9–17]. However, the high $\chi$-low $N$ BCPs are synthesized partially or entirely by polymerization reactions. As a result, it is virtually impossible for all growing polymer chains to terminate at the same length; therefore, the BCPs exhibit a range of molecular-weight distributions (typically, dispersity of 1.05–1.30)[9–16]. Such a dispersity is particularly detrimental for low-molecular-weight BCPs targeting a domain spacing of few nanometers (i.e., $d \approx 5$ nm) since at that scale, a single monomer unit size rapidly approaches $d$. In other words, in such low-molecular-weight regimes, increasing or decreasing the number of BCP monomer units by only a single unit would impact the resulting self-assembly structural properties, such as the morphology, $d$, defects, roughness, and ordering[18–20]. Indeed, seminal works by Mejier et al.[21] and Bates and Hawker et al.[22] have demonstrated that $d$ increased with increasing BCP dispersity. More importantly, in the low-molecular-weight (~2500 g mol⁻¹) region, increasing the dispersity resulted in a loss of microphase separation, despite the fact that the monodisperse counterparts could form well-ordered nanostructures[23]. Another problem with conventional BCPs is synthesis reproducibility, whereby it is virtually impossible to reproduce BCPs showing exactly the same molecular weights and compositions, thereby causing batch-dependent variation in the resulting nanostructures. Given the importance of monodispersity in constructing ultrasmall nanostructures, a paradigm shift from conventional polydisperse BCPs to monodisperse block co-oligomer (BCO) systems is highly desirable.

Although several monodisperse BCP/BCO systems have been reported, their syntheses rely on either chromatographic fractionation of polydisperse polymers[22] or iterative reactions to grow monomer units in a step-by-step manner[21,23–25], both of which require laborious efforts and time-consuming processes, thereby posing a major hurdle to meaningful progress in the fundamental science of monodisperse BCPs/BCOs. Therefore, the preparation of a readily available monodisperse BCO system that can form various microphase-separated morphologies showing ultrasmall periodicity is particularly challenging.

To develop such ideal BCO systems, we herein propose a combination of sugars and terpenoids. The previous studies into oligosaccharide-based BCPs synthesized via the "click" chemistry revealed the formation of small microphase-separated structures[26–30] with the $d$ of even down to 5.8 nm (in the bulk state)[31,32]. Therefore, strong segregation between hydrophilic saccharidic blocks and hydrophobic isoprene-based hydrocarbon chains would be expected, which would in turn allow microphase separation in the molecular-weight regime below 2000 g mol⁻¹. In addition, recent reports[33–35] by Lodge, Siepmann, and Hillmyer on sugar-based amphiphiles with a hydrocarbon chain, leading to a lamellar morphology with $d$ of 3.5 nm (even down to $d$ of 1.2 nm

as revealed by molecular dynamic simulation), also supports that the combination of sugars and terpenoids is highly promising for realizing ultrasmall nanostructure formation. More importantly, oligosaccharide and terpenoid blocks showing defined and discrete degrees of polymerization (DPs) are commercially available, and a series of BCOs featuring monodispersity and discrete DPs can be synthesized readily from them, providing rapid access to various ultrasmall nanostructures showing diverse morphologies, features, and sizes. Thus, we herein employ a series of sugars from monosaccharide (i.e., glucose) to heptasaccharide (i.e., maltoheptaose) as oligosaccharide blocks, while farnesol (3,7,11-trimethyldodeca-2,6,10-trien-1-ol; mixture of isomers), DL-$\alpha$-tocopherol (2,5,7,8-tetramethyl-2-[(4,8,12-trimethyltridecyl)]chroman-6-ol), and solanesol ((2E,6E,10E,14E,18E,22E,26E,30E)-3,7,11,15,19,23,27,31,35-nonamethyl-hexatriaconta-2,6,10,14,18,22,26,30,34-nonaen-1-ol)—which are a family of terpenoids containing three, four, and nine isoprene units, respectively—are selected as hydrophobic blocks. Using this BCO system, we attempt the construction of LAM, GYR, and hexagonally close-packed cylindrical (HEX) morphologies by adjusting the DP of each block, and the $d$ values of the resulting organic-BCP/BCO-based microphase-separated structures are examined.

## Results

**Synthesis**. Our BCO synthetic procedure relies on the "click" reaction between terpenoid (i.e., farnesol, tocopherol, and solanesol) and sugar units, as depicted in Fig. 1. Solanesol-based BCOs ($Glc_n$-$b$-Sol; $n = 1$–7) were synthesized by the "click" reactions of the azido-functionalized solanesol ($N_3$-Sol) with propargyl-functionalized glucose ($Glc_1$-C≡CH), maltose ($Glc_2$-C≡CH), maltotriose ($Glc_3$-C≡CH), maltotetraose ($Glc_4$-C≡CH), maltopentaose ($Glc_5$-C≡CH), maltohexaose ($Glc_6$-C≡CH), and maltoheptaose ($Glc_7$-C≡CH) in the presence of a Cu catalyst in dimethylformamide (DMF) and tetrahydrofuran (THF) mixed solvent. Following the completion of the reaction, the Cu catalyst was removed by treating the reaction mixture with a cation-exchange resin, and the resulting residue was precipitated in a selective solvent to remove unreacted starting materials, thereby affording the desired BCOs in moderate to good yields. The reaction products did not exhibit any strong infrared (IR) absorption at 2100 cm⁻¹, suggesting near-quantitative consumption of the azido group (Supplementary Fig. 1). Proton nuclear magnetic resonance (¹H NMR) spectra were assigned to the expected BCO chemical structure (Supplementary Figs. 2–8), and the size-exclusion-chromatography (SEC; in DMF containing 0.01 M LiCl) elution peak clearly shifted toward the higher-molecular-weight region upon increasing the number of glucose units in the BCO (Fig. 2a). More importantly, matrix-assisted laser desorption/ionization–time-of-flight mass spectrometry (MALDI-TOF MS) analysis confirmed the presence of mono-disperse and discrete products (Fig. 2b and Supplementary Fig. 9). It should be emphasized that successful multigram-scale synthesis could be attributed to the facile preparation of the starting materials and simple purification. Similarly, tocopherol- and farnesol-based BCOs ($Glc_1$-$b$-Toc, $Glc_2$-$b$-Toc, and $Glc_1$-$b$-Far) were prepared and characterized (Supplementary Figs. 9–15).

**Nanostructural formation of solanesol-based BCOs**. To gather fundamental information regarding the microphase separation behavior, a series of thermally annealed solanesol-based BCOs ($Glc_n$-$b$-Sol; $n = 1$–7) was subjected to small-angle X-ray scattering (SAXS) (Fig. 3a). Prior to the SAXS experiment, we performed differential scanning calorimetry (DSC) analysis on the BCOs. Although all the BCO DSC thermograms did not exhibit

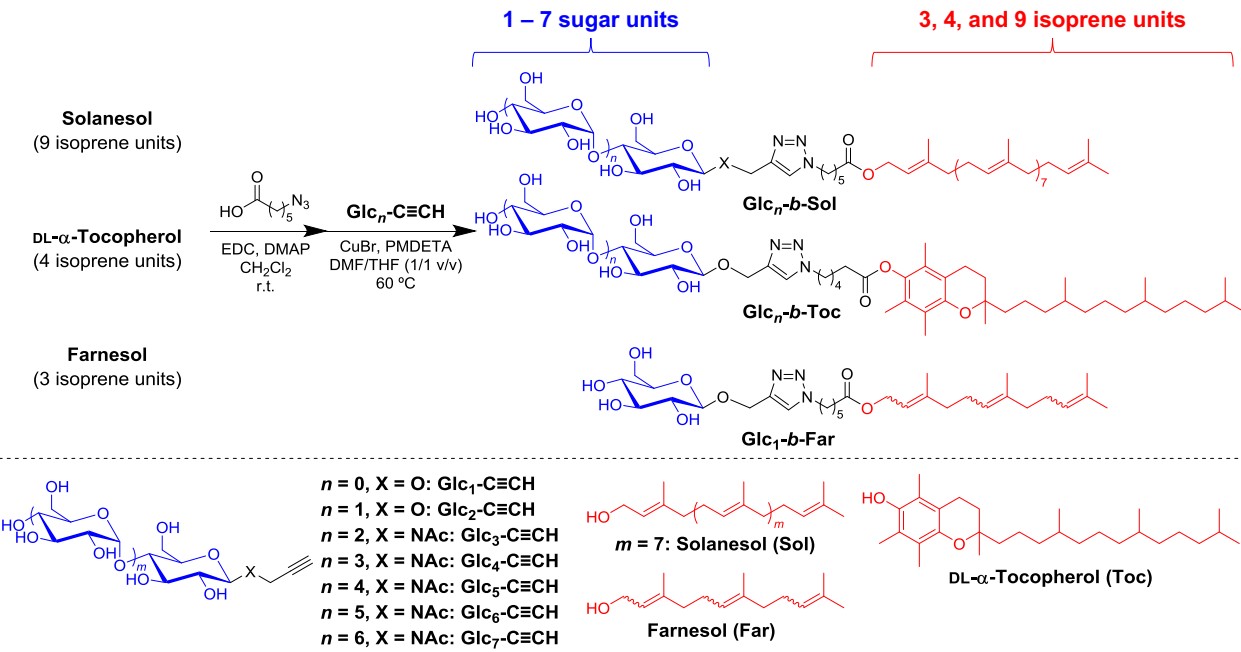

**Fig. 1 Synthetic schemes.** Preparation of the monodisperse and discrete $Glc_n$-$b$-Sol ($n = 1$–7), and $Glc_n$-$b$-Toc ($n = 1$ and 2), and $Glc_1$-$b$-Far.

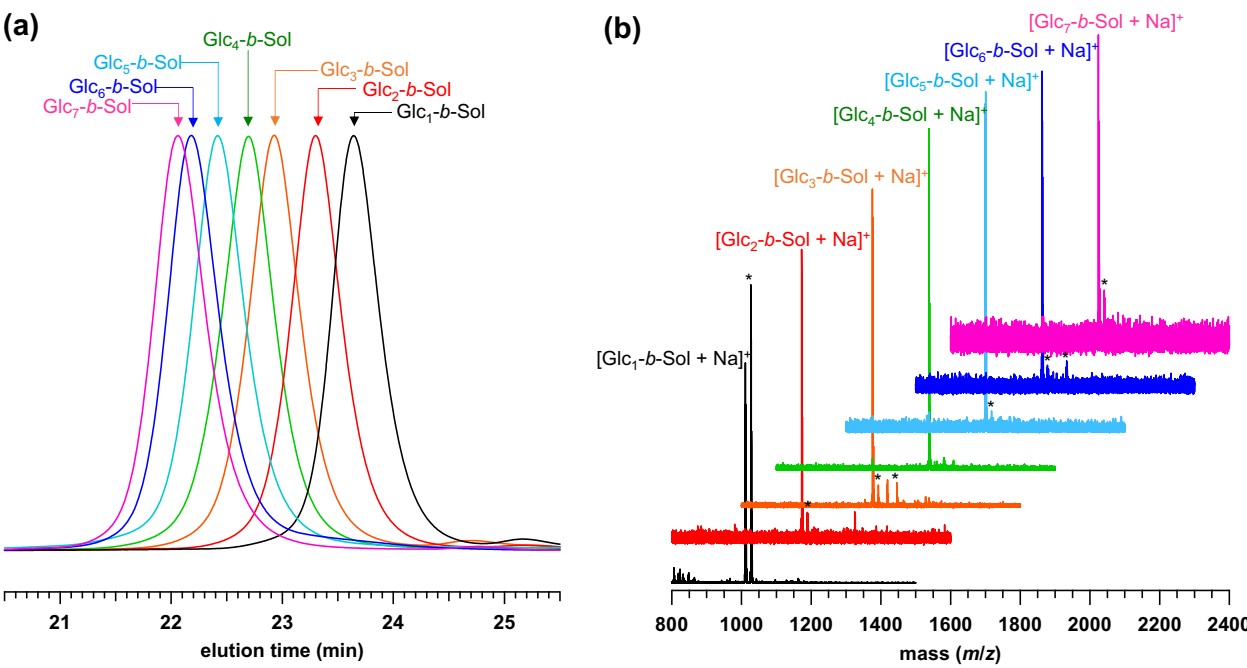

**Fig. 2 Characterization of $Glc_n$-$b$-Sol. a** SEC traces (in DMF containing 0.01 M LiCl) and **b** MALDI-TOF mass spectra for $Glc_n$-$b$-Sol ($n = 1$–7). Asterisks indicate peaks attributed to $[M+K]^+$ or $[M+Ag]^+$.

any transitions due to crystallization and melting during the first cooling and second heating processes, they did exhibit two baseline shifts corresponding to glass transition temperatures ($T_g$s) for the solanesol and sugar segments (Supplementary Fig. 16). The fact that the $T_g$s of the two segments were independently observed suggested the microphase separation between the segments.

Unless otherwise noted, bulk samples were annealed at 130 °C for 36 h and then quenched to room temperature. The SAXS profiles of $Glc_1$-$b$-Sol, $Glc_2$-$b$-Sol, $Glc_3$-$b$-Sol, $Glc_4$-$b$-Sol, and $Glc_5$-$b$-Sol showed higher-order scattering peaks and a sharp

primary peak ($q^*$), demonstrating the formation of well-ordered periodic nanostructures (Table 1). It is rather surprising that even the BCO containing only one glucose unit, i.e., $Glc_1$-$b$-Sol, had self-assembled into the HEX morphology with a domain spacing ($d = 2\pi/q^*$) of 5.5 nm (cylinder-to-cylinder distance $= 2d/\sqrt{3} = 6.4$ nm; cylinder diameter $= [8f_{sugar} \cdot d^2/(\sqrt{3} \cdot \pi)]^{1/2} = 2.7$ nm, where $f_{sugar}$ is the saccharide volume fraction calculated based on the densities of 1.36 and 0.95 for the saccharide and solanesol blocks, respectively)[23], which is among the smallest ever obtained among structures that have been self-assembled from saccharide-based block BCPs. For example, $d$ values of 5.8 and 6.6 nm were

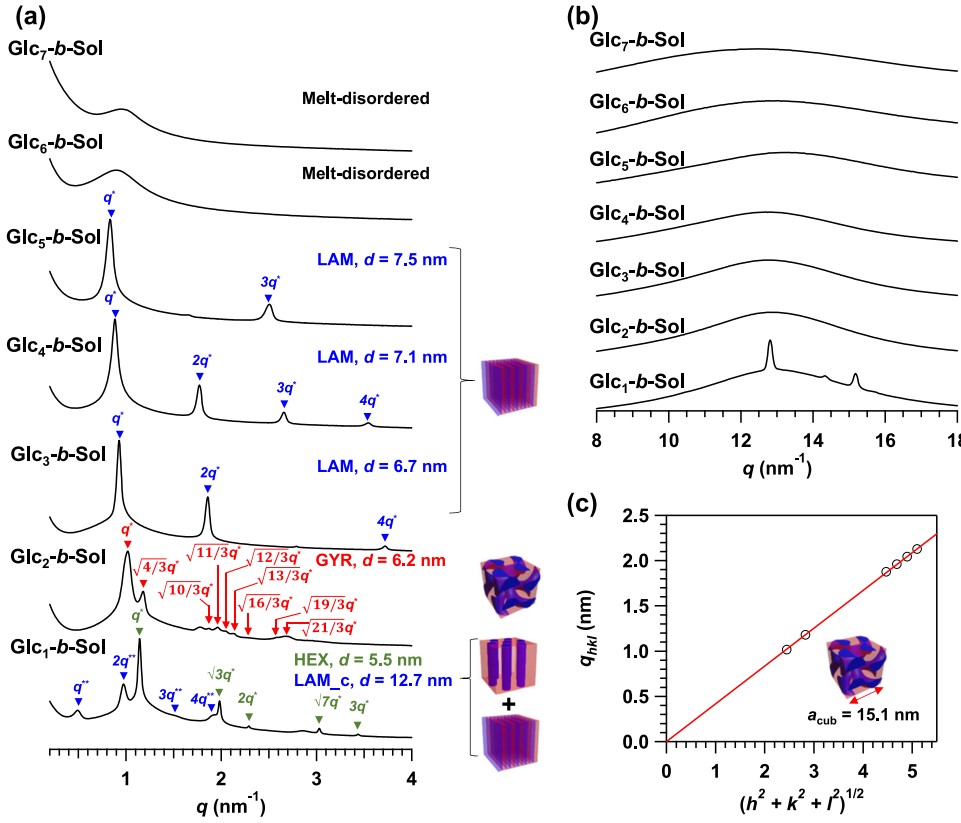

**Fig. 3 Morphological analyses for the thermally annealed bulk Glc$_n$-$b$-Sol ($n$ = 1–7) samples. a** SAXS profiles revealed the formation of ordered nanostructures, with the exceptions of Glc$_6$-$b$-Sol and Glc$_7$-$b$-Sol. **b** WAXS profiles of Glc$_n$-$b$-Sol ($n$ = 1–7) demonstrating amorphous BCOs, with the exception of Glc$_1$-$b$-Sol. Glc$_2$-$b$-Sol was annealed at 100 °C for 36 h while all other samples were annealed at 130 °C for 36 h. **c** $q_{hkl}$ versus ($h^2 + k^2 + l^2$)$^{1/2}$ plot based on the SAXS pattern for Glc$_2$-$b$-Sol validated space group assignment, where $h$, $k$, and $l$ are the Miller indices of the corresponding planes. The lattice parameter $a_{cub}$ was calculated from the slope of the linear fit.

**Table 1 Molecular parameters and bulk morphological characteristics of Glc$_n$-$b$-Sol ($n$ = 1–7), Glc$_1$-$b$-Far, and Glc$_n$-$b$-Toc ($n$ = 1 and 2).**

| BCO | DP$_{Glc}$[a] | DP$_{isoprene}$[a] | M.W. | $f_{sugar}$[b] | Bulk morphology[c] (annealing condition) | $d$[c] (nm) |
|---|---|---|---|---|---|---|
| Glc$_1$-$b$-Sol | 1 | 9 | 988.44 | 0.16 | HEX/LAM_c (130 °C, 36 h) | 5.5/12.7 |
| Glc$_2$-$b$-Sol | 2 | 9 | 1150.58 | 0.25 | GYR (100 °C, 36 h) | 6.2 |
| Glc$_3$-$b$-Sol | 3 | 9 | 1353.78 | 0.33 | LAM (130 °C, 36 h) | 6.7 |
| Glc$_4$-$b$-Sol | 4 | 9 | 1515.92 | 0.39 | LAM (130 °C, 36 h) | 7.1 |
| Glc$_5$-$b$-Sol | 5 | 9 | 1678.06 | 0.44 | LAM (130 °C, 36 h) | 7.5 |
| Glc$_6$-$b$-Sol | 6 | 9 | 1840.2 | 0.48 | Melt-disordered | – |
| Glc$_7$-$b$-Sol | 7 | 9 | 2002.34 | 0.52 | Melt-disordered | – |
| Glc$_1$-$b$-Toc | 1 | 4 | 802.09 | 0.21 | GYR (80 °C, 6 h) | 5.2 |
| Glc$_2$-$b$-Toc | 2 | 4 | 964.23 | 0.31 | LAM (80 °C, 6 h) | 6.3 |
| Glc$_1$-$b$-Far | 1 | 3 | 579.72 | 0.28 | LAM (80 °C, 6 h) | 4.2 |

[a]DPs BCO glucose and isoprene units (DP$_{Glc}$ and DP$_{isoprene}$, respectively).
[b]The saccharide volume fraction ($f_{sugar}$) was calculated based on densities of 1.36, 0.90, and 0.95 g cm$^{-3}$ for the saccharide, solanesol, and farnesol, and tocopherol blocks, respectively.
[c]The morphology and $d$ were determined from bulk-sample SAXS data.

reported for LAM-forming cellobiose-*block*-polypropylene[31] and HEX-forming maltotriose-*block*-polycaprolactone systems[36], respectively, in the bulk states. In addition, a LAM morphology showing relatively large $d$ = 12.7 nm had formed through self-assembly. Considering the multiple diffraction peaks present in the wide-angle X-ray scattering (WAXS) profiles (Fig. 3b), the coexisting LAM morphology should be a crystallization-driven rather than a microphase-separated structure (Supplementary Figs. 17 and 18). We thus denote this nanostructure as LAM_c to differentiate with the microphase-separated LAM structure.

Indeed, in situ SAXS measurements obtained while heating the samples gave only the scattering pattern characteristic of HEX ($q/q^*$ = 1, √3, √4, and √7; Supplementary Fig. 18), confirming the absence of the LAM morphology at elevated temperatures.

The SAXS profile of Glc$_2$-$b$-Sol thermally annealed at 100 °C for 36 h showed scattering peaks at $q$ = 1.017, 1.179, 1.875, 1.961, 2.042, and 2.128 nm$^{-1}$, which were assignable to the (211), (220), (420), (332), (422), and (431) crystallographic planes of the GYR phase, respectively (Fig. 3a)[37]. Although a similar scattering pattern assignable to the GYR phase was found in the sample

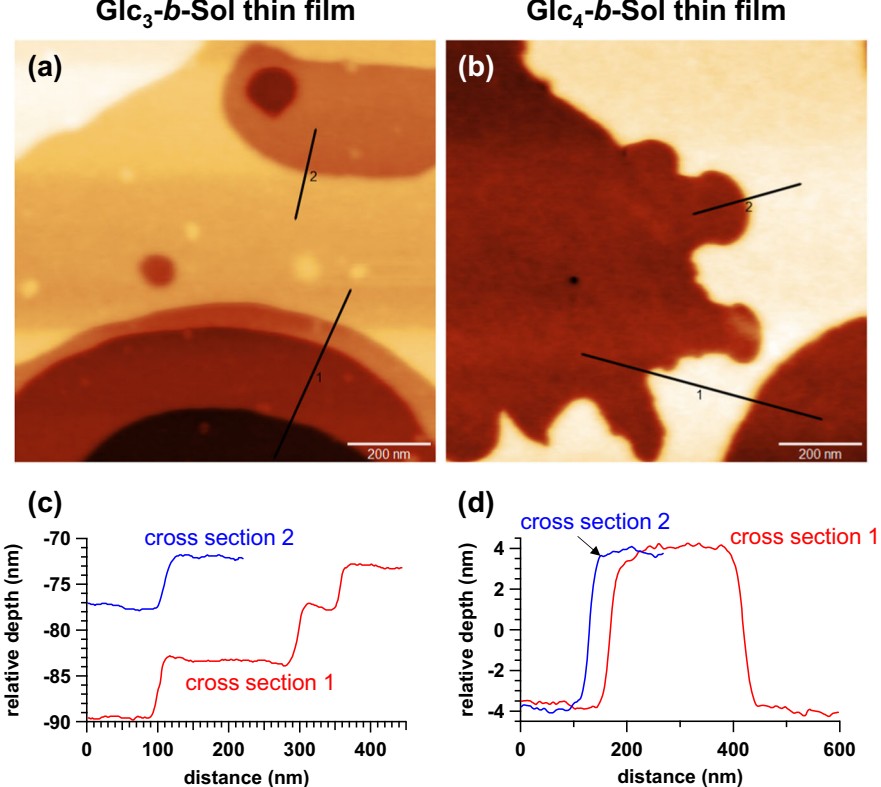

**Fig. 4 Thin-film morphologies of Glc$_3$-*b*-Sol and Glc$_4$-*b*-Sol.** AFM height images (**a**, **b**) and corresponding cross-sectional profiles (**c**, **d**) indicating the formation of 6–8-nm-thick horizontal lamellae in Glc$_3$-*b*-Sol (**a**, **c**) and Glc$_4$-*b*-Sol thin films (**b**, **d**). Thin-film samples were prepared by spin-coating the BCO solution onto the hydrophilic surface of a silicon substrate followed by thermal annealing at 85 °C for 1 h.

after 130 °C for 36 h, the scattering peak width was broader, suggesting less ordering (Supplementary Fig. 19). Note that this is the first-ever example of GYR-forming saccharide-based BCPs. Since the maltose block is the minor component, the interwoven network and surrounding matrix should be composed of maltose and solanesol, respectively. The cubic lattice parameter $a_{cub}$ was determined to be 15.1 nm from the slope of a plot of $q_{hkl}$ versus $(h^2 + k^2 + l^2)^{1/2}$ ($a_{cub} = 2\pi(h^2 + k^2 + l^2)^{1/2}/q_{hkl}$), where $h$, $k$, and $l$ are Miller indices (Fig. 3c). The lateral distance between interwoven networks was then calculated to be 6.5 nm ($=\sqrt{3}a_{cub}/4$)[38], which is approximately the fully extended molecular length (ca. 7 nm). Thus, low BCO molecular weight should be responsible for the GYR nanomorphology.

SAXS profiles were then generated for the LAM structures showing $q/q^*$ ratios of 1:2:3:4 after increasing the number of glucose units to more than 3 (i.e., for Glc$_3$-*b*-Sol, Glc$_4$-*b*-Sol, and Glc$_5$-*b*-Sol) (Fig. 3a); $d$ was in the range 6.7–7.5 nm and increased in ca. 0.4 nm increments with each glucose unit added. It should be notable that Glc$_3$-*b*-Sol formed the LAM despite the asymmetric composition ($f_{sugar} = 0.33$). According to classical BCP theory, a volume fraction of ~0.3 should lead to HEX rather than LAM structural formations[39]. However, the rigid monodisperse oligosaccharide molecules encumbered BCO molecules from packing into higher-curvature morphologies, thus preferentially forming LAM structures even at such asymmetric volume ratios. For LAM structure, $d$ is the sum of the lamellar microdomain thicknesses for the two constitutional blocks. Given the 1:1 volume ratio of Glc$_5$-*b*-Sol as revealed by the SAXS analysis, the lamellar microdomain thickness for each block should be ~3.8 nm.

Further increasing the number of glucose units resulted in poor ordering (i.e., random two phase) in the bulk (Fig. 3a). Despite

thermally annealing Glc$_6$-*b*-Sol and Glc$_7$-*b*-Sol up to 180 °C for 36 h, their SAXS profiles only exhibited a single broad peak without any higher-order scattering. The reduced chain mobility associated with high glass transition temperatures ($T_g$s) owing to longer oligosaccharide molecules (e.g., 145 and 150 °C for maltohexaose and maltoheptaose, respectively[40]) should be responsible for restricting the self-assembly process.

BCO thin films were then prepared by spin-coating a toluene/DMF BCO solution onto silicon substrates. To confirm the ordered nanostructures in the BCO thin films, we investigated Glc$_n$-*b*-Sol ($n$ = 1–5) thin films by grazing-incidence small-angle X-ray scattering (GISAXS) measurements and atomic force microscopy (AFM). GISAXS analysis revealed that the thin films presented horizontally orientated LAM morphologies, regardless of the number of glucose units (Supplementary Fig. 20). In addition, the AFM images of the Glc$_3$-*b*-Sol and Glc$_4$-*b*-Sol thin films clearly showed terrace formations (Fig. 4), which agreed with the horizontally orientated LAM formations observed by GISAXS. The height difference between two consecutive layers (~6 and ~8 nm for Glc$_3$-*b*-Sol and Glc$_4$-*b*-Sol, respectively) roughly matched $d$, as determined by SAXS and listed in Table 1. The key morphological features of the Glc$_n$-*b*-Sol thin films were consistent with horizontal orientations retained over large areas and relatively low distributions of LAM thickness. Such structural uniformity should be a direct result of the BCO monodispersity.

**Fine-tuning nanostructures by varying dispersity.** The most important feature of the present BCO system is its monodispersity, which should allow high degrees of ordering in the bulk. Previous studies have suggested the possibility of tailoring BCP microphase separation—including size, morphology, and

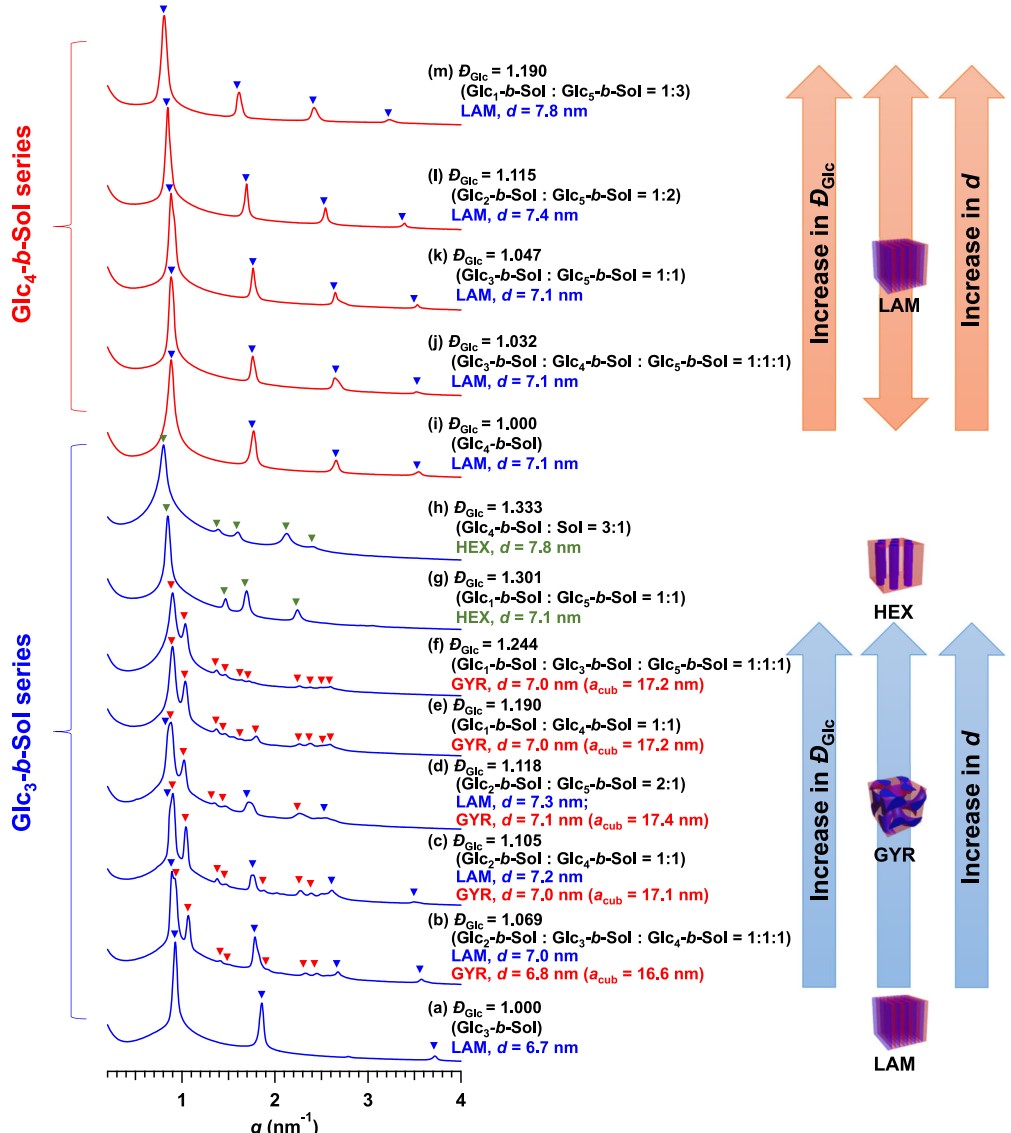

**Fig. 5 Effect of dispersity on Glc$_3$-$b$-Sol and Glc$_4$-$b$-Sol microphase separations.** SAXS profiles of the polydisperse (**a–h**) Glc$_3$-$b$-Sol and (**i–m**) Glc$_4$-$b$-Sol series revealed variation in the morphology and $d$ upon increasing Đ$_{Glc}$. Polydisperse samples were prepared by mixing different BCO molecules in given molar ratios. All samples were thermally annealed at 130 °C for 36 h. Blue, red, and green inverted triangles indicate peak assignments corresponding to the LAM, GYR, and HEX structures, respectively.

stability—by tuning the constitutional-block dispersities[18,41–45]. Thus, we hypothesized that BCO mixtures with different numbers of glucose units could further expand the morphologies, features, and sizes accessible from the current BCO system. To test this hypothesis, we investigated the effects of sugar-block dispersity (Đ$_{Glc}$) on microphase-separated structures. By mixing different Glc$_n$-$b$-Sol molecules ($n = 1–5$), polydisperse versions of Glc$_3$-$b$-Sol and Glc$_4$-$b$-Sol were artificially prepared through variation in Đ$_{Glc}$ while fixing the average number of glucose units. For example, Glc$_1$-$b$-Sol and Glc$_5$-$b$-Sol mixed in a 1:1 molar ratio showed the same average number of glucose units as Glc$_3$-$b$-Sol but a different Đ$_{Glc}$. All thermally annealed polydisperse samples underwent self-assembly into well-ordered microphase-separated structures regardless of Đ$_{Glc}$, as evidenced by the multiple sharp scattering peaks in their SAXS profiles (Fig. 5). Compared to the microphase-separated morphology of monodisperse Glc$_3$-$b$-Sol, that of polydisperse Glc$_3$-$b$-Sol shifted from LAM (Đ$_{Glc}$ = 1.000) to GYR (Đ$_{Glc}$ = 1.190 and 1.244) and finally to HEX (Đ$_{Glc}$ = 1.301 and 1.333) with increasing Đ$_{Glc}$ (Fig. 5a–h, see

Supplementary Fig. 21 for the GYR peak assignment). The coexistence of longer and shorter oligosaccharides facilitated the formation of curved microdomains, thereby allowing the morphology to shift from a lower to a higher curvature[18]. In addition to the morphological shift, $d$ increased with increasing Đ$_{Glc}$. Interestingly, for the GYR-forming samples, the full width at half maximum (Δ$q^*$) of the primary scattering peak increased with increasing Đ$_{Glc}$ (e.g., Δ$q^*$ = 0.0326 nm$^{-1}$ for Đ$_{Glc}$ = 1.069; Δ$q^*$ = 0.0365 nm$^{-1}$ for Đ$_{Glc}$ = 1.244). Since the Δ$q^*$ is inversely proportional to the grain size according to the Scherrer's equation[46], this tendency implies the decrease in the GYR grain size with increasing Đ$_{Glc}$. In the Glc$_4$-$b$-Sol series, on the other hand, $d$ increased from 7.1 nm (Đ$_{Glc}$ = 1.000) to 7.8 nm (Đ$_{Glc}$ = 1.165) with increasing Đ$_{Glc}$, while the LAM morphology remained unchanged (Fig. 5i–m). Glc$_4$-$b$-Sol shows a higher saccharide volume fraction ($f_{sugar}$) than Glc$_3$-$b$-Sol, and therefore, is deeply embedded in the LAM region of the phase diagram. This likely accounts for the fact that the Glc$_4$-$b$-Sol series did not exhibit a morphological shift. In contrast, Glc$_3$-$b$-Sol is likely located near

the LAM–GYR phase boundary; therefore, slight variations in dispersity shifted the resulting morphology. Since the morphology and $d$ value both correlated with $Đ_{Glc}$, the morphology can be tailored, and $d$ can be adjusted by a few angstroms simply by blending discrete BCO molecules. Conversely, however, these results imply that monodisperse and discrete natures are essential in fabricating highly reproducible ultrasmall nanostructures, which is particularly important in the context of nanolithographic applications.

**Further minimizing the feature size by tocopherol- and farnesol-based BCOs.** Although solanesol-based BCOs underwent self-assembly into highly ordered nanostructures, our obtained results indicated that the lower limit of $d$ achievable when combining solanesol with the monosaccharide was ~6 nm. We, therefore, examined the BCO nanostructures formed from shorter terpenoid blocks (i.e., Glc$_1$-$b$-Toc, Glc$_2$-$b$-Toc, and Glc$_1$-$b$-Far) by SAXS analysis. Surprisingly, the SAXS profiles of the thermally annealed Glc$_1$-$b$-Toc, Glc$_2$-$b$-Toc, and Glc$_1$-$b$-Far (80 °C for 6 h) exhibited multiple scattering peaks corresponding to GYR ($d = 5.2$ nm; $a_{cub} = 12.6$ nm), LAM ($d = 6.3$ nm), and LAM ($d = 4.2$ nm) nanostructures, respectively (Fig. 6a and Supplementary Fig. 22), despite their extremely low M.W. (580–964 g mol$^{-1}$) and total DP (4–6) (Table 1). Such an ultrasmall $d$ value of ~4 nm has rarely been reported among the organic-BCP-derived LAM microphase-separated structures[9,14,23,31,34,47]. Given the $f_{sugar}$ of Glc$_1$-$b$-Far, the sugar microdomain thickness is approximately 1.2 nm, which shows promise to ultrahigh-resolution nanofabrication. In addition, we spin-coated a Glc$_1$-$b$-Toc thin film, and the corresponding GISAXS image showed a strong scattering spot along the out-of-plane direction, indicating the formation of a horizontally orientated LAM morphology (Supplementary Fig. 23). Furthermore, the corresponding AFM height image clearly shows terrace formation corresponding to multilayer LAM stacking, which also supports the GISAXS result (Fig. 6b). The height difference between two consecutive layers (i.e., $d$ for the LAM morphology) was ~4 nm (Fig. 6c), which again confirmed the astonishing capability of the present BCO system to form ultrasmall nanostructures.

**Phase behavior of sugar/terpenoid BCO Systems.** One of the most important findings of our study is that the resulting BCO morphology varies from HEX to GYR and finally to LAM with increasing $f_{sugar}$, as summarized in the conceptual phase diagram (Fig. 7). Note that we tentatively employed the total number of glucose and isoprene units ($DP_{Glc} + DP_{isoprene}$) as the phase-diagram vertical axis because it is highly challenging to determine $\chi$ and $N$ for the present BCO system. The observed propensity is consistent with the conventional BCP microphase separation theory[39]. In other words, the desired morphology and feature size can both be achieved by simply designing the number of BCO glucose and isoprene units, which is analogous to the molecular design of conventional BCPs based on $f$ and $N$. Unfortunately, however, a spherical morphology could not be obtained from the current combination of oligosaccharides and terpenoids. The fact that the most asymmetric Glc$_1$-$b$-Sol formed a cylindrical morphology suggests the difficulty in obtaining spherical morphologies by the current molecular design.

**Conclusion**
We herein reported the development of a readily available monodisperse and discrete block co-oligomer (BCO) system consisting of hydrophilic sugars and hydrophobic terpenoids. Starting from readily available natural sugars and terpenoids with defined numbers of glucose and isoprene units, BCOs exhibiting

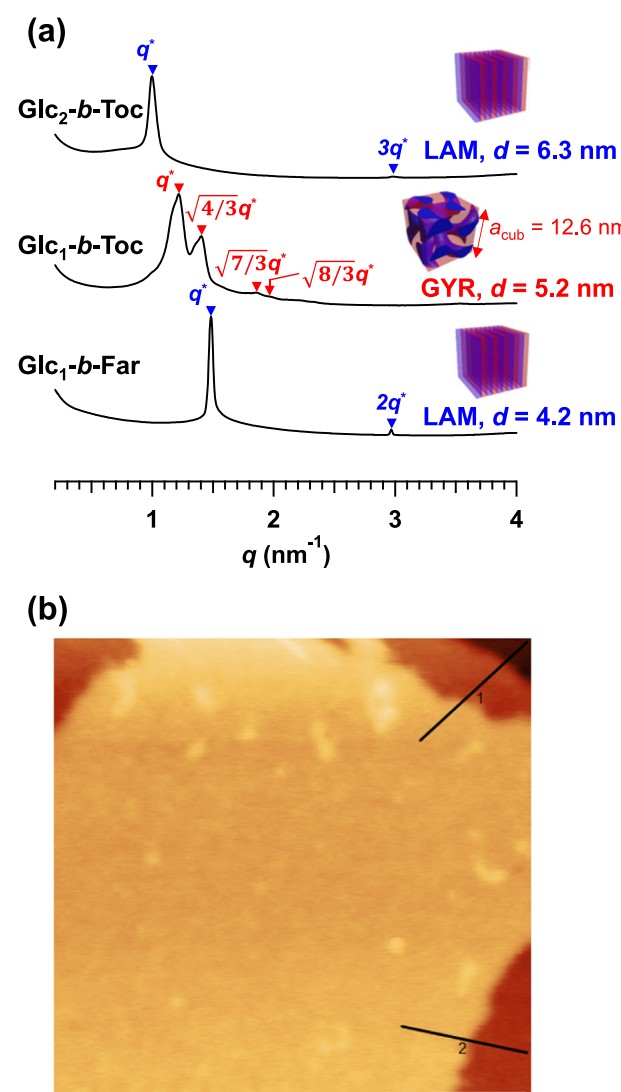

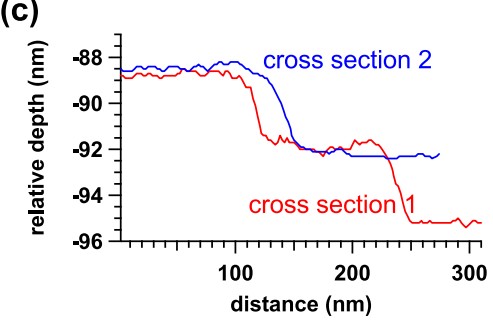

**Fig. 6 Morphological analyses of Glc$_1$-$b$-Toc, Glc$_2$-$b$-Toc, and Glc$_1$-$b$-Far. a** SAXS profiles for the bulk samples thermally annealed at 80 °C for 6 h revealed $d = 4$–6 nm microphase separation. **b** AFM height image, and **c** corresponding cross-sectional profiles of the as-cast Glc$_1$-$b$-Far thin film confirmed the horizontally orientated LAM formation in ~4-nm-thick layers.

the desired total degrees of polymerization (DPs) and compositions could be synthesized easily by connecting these blocks via a "click" reaction. The resulting BCOs self-assembled into well-ordered lamellar (LAM), gyroid (GYR), and hexagonally close-packed cylindrical (HEX) nanostructures showing 4–8 nm periodicity through simple thermal annealing. The microphase-

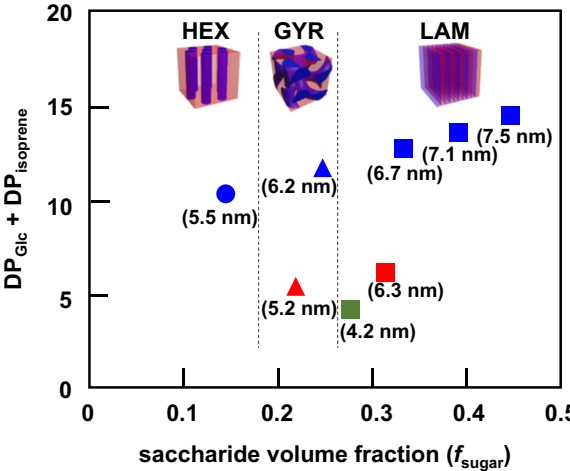

**Fig. 7 Conceptual phase diagram.** Conceptual phase diagram was generated for the phase-separated Glc$_n$-b-Sol ($n = 1$–5; blue), Glc$_n$-b-Toc ($n = 1$ and 2; red), and Glc$_1$-b-Far (green) bulk morphologies. Values in parentheses are the $d$ values listed in Table 1. The vertical and horizontal axes represent the sum of glucose and isoprene DPs and the saccharide volume fraction, respectively.

separated morphology shifted from cylindrical to GYR and finally to LAM upon increasing the number of glucose units (i.e., the saccharide volume fraction) for a fixed hydrophobic segment length, which is analogous to well-established BCP phase behavior. Therefore, the desired BCO morphologies, features, and sizes can be achieved simply by designing the number of BCO glucose and isoprene units appropriately. Most importantly, we developed a well-ordered microphase-separated structure showing 4.2 nm periodicity and 1.2 nm microdomain thickness, which is among the smallest ever achieved by organic block copolymer (BCP)/BCO-based microphase-separated structures. Considering the commercial availability of all starting materials, the ease of BCO synthesis, and the facile BCO purification, we believe that presented BCOs strongly contribute to accelerating applied research of solid and solution state self-assembly of discrete and monodisperse BCOs, which will inevitably expand their application scopes in various fields of not only the nanolithography but also organic devices, separation materials, coatings, etc. We are currently working on the orientational control of the BCO nanostructures in the thin-film state to achieve nanolithography with a few-nanometer resolution.

## Methods

**Synthesis and characterization.** See Supplementary Methods and Supplementary Figs. 1–15.

**Typical synthesis procedure for Glc$_n$-b-Sol.** N$_3$-Sol (6.00 g, 7.79 mmol), Glc$_2$-C≡CH (3.80 g, 9.35 mmol), and copper(I) bromide (CuBr; 112 mg, 779 μmol) were placed into a Schlenk flask, which was evacuated and back-filled with argon three times. A solution of N,N,N′,N″,N″-pentamethyldiethylenetriamine (134 mg, 779 mmol) in a mixed solvent of DMF (25.0 mL) and THF (25.0 mL) was degassed by the argon bubbling, and the mixture was then transferred into the Schlenk flask under an argon atmosphere. The reaction mixture was stirred at 60 °C for 72 h and then treated with Dowex to remove Cu catalyst. The product was purified by reprecipitation from the DMF solution into a mixed solvent of acetonitrile and H$_2$O (7/3 (v/v)) to give Glc$_2$-b-Sol as a white solid (6.29 g, yield: 70.2%).

**Small-angle X-ray scattering (SAXS) experiments.** SAXS experiments for bulk samples were performed at BL-6A beamline of the Photon Factory of High Energy Accelerator Research Organization (KEK, Tsukuba, Japan). The X-ray wavelength and exposure time were 1.50 Å (8.27 keV) and 60 s, respectively. A PILATUS3 1M (Dectris Ltd., Switzerland) detector, with 981 × 1043 pixels at a pixel size of 172 × 172 μm, and a counter depth of 20 bits (1,048,576 counts), was used for data

acquisition. The sample-to-detector distance was calibrated using the scattering patterns of silver behenate. The bulk samples were put into 1.5 mm diameter glass capillaries. After that, the bulk samples were annealed under vacuum at 70, 100, and 130 °C for 36 h. The SAXS data were acquired under ambient condition, and 1D profiles were obtained as plots of scattering intensity as functions of scattering vector ($q$), where $q = (4\pi/\lambda) \sin(\theta/2)$ ($\lambda$, wavelength; $\theta$, scattering angle).

**Atomic force microscopy (AFM).** The AFM phase images were realized using a molecular imaging PicoPlus atomic force microscope operating in the tapping mode with a silicon cantilever (PPP-NCHR and SSS-NCHR, NANOSENSORS) having a resonant frequency of 330 kHz and a force constant of 42 N m$^{-1}$. The AFM images were processed using Gwyddion software.

## Data availability

The authors declare that the data supporting the findings of this study are available within the article and Supplementary Information file, including experimental procedures and characterization data for all the new compounds.

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

## Acknowledgements

This work was financially supported by a JSPS Grant-in-Aid for Scientific Research (B) (No. 19H02769, T.S.; No. 20H02792, T.I.), a JSPS Grant-in-Aid for Young Scientists (No. 18K14268, T.I.), the Photoexcitonix Project (Hokkaido University, T.S.), the Frontier Chemistry Center (Hokkaido University, T.S. and T.I.), the Creative Research Institute (CRIS; Hokkaido University, T.S.), the Tonen General Sekiyu Research/ Development Encouragement & Scholarship Foundation (T.I.), the Shorai Foundation for Science and Technology (T.I.), the Iketani Science and Technology Foundation (T.I.), and the Asahi Glass Foundation (T.I.). This work was performed in part under the approval of the Photon Factory Program Advisory Committee (Proposals Nos. 2017G589 and 2019G579). T.I. gratefully acknowledges the Nanotech CUPAL NRP program. B.J.R., K.W., and K.Y. were funded by the JSPS Fellowship for Young Scientists. We thank Prof. Hajime Ito (Hokkaido University) for his assistance with AFM experiments.

## Author contributions

T.I. and T.S. designed the experiment. T.I. wrote the manuscript. T.I., R.K., C.L., and N.K. synthesized and characterized the BCOs and analyzed the X-ray data. B.J.R., K.W., K.Y., and M.H. helped in X-ray measurements and analysis. K.W., K.Y., and K.T. helped in AFM measurements. R.B. and T.Y. contributed to the discussion. The manuscript was written through the contributions of all authors. All authors have given their approval to the final version of the manuscript.

## Competing interests

The authors declare no competing interests.
