## [Peer Review File · Communications Chemistry]

Reviewers' comments:

Reviewer #1 (Remarks to the Author):

In this paper, a block co-oligomer (BCO) system consisting of hydrophilic sugars and hydrophobic terpenoids was synthesized by click reaction. Under thermal annealing, the resulting BCOs self-assembled into well-ordered lamellar (LAM), gyroid (GYR), and hexagonally close-packed cylindrical (HEX) nanostructures showing 4-8 nm periodicity, which are among smallest domains ever reported. These BCO nanostructures show various application potentials in nanolithography, organic devices, separation materials, coatings, and other fields.

The discovery of the self-assembled domain with a minimal 4.2 nm periodicity is appealing to the current block copolymer lithography research field. Besides, the tailoring on self-assembled morphology by changing the number of glucose or isoprene units, and tuning on Δ Glc in BCO is convincing and comprehensive. Therefore, this paper will be published in communications chemistry with minor revisions.

I have several questions as follows:

1. In Table 1, the author should explain why the annealing condition for preparation GYR morphology (100 °C, 36 h) is different from the others (130 °C, 36 h). Perhaps the GYR morphology is in a meta-stable state and a morphology transition may occur at the higher annealing temperature.
2. The author reported a few GYR morphologies for the studied BCO. However, the morphology analysis is merely based on the detected peak ratios in the SAXS profile. To make the discovery more solid, TEM characterization is suggested, as ruthenium tetroxide may selectively stain the hydroxyl groups-rich sugar-based segments and osmium tetroxide may selectively stain the isoprene segments. Therefore, the contrast between different domains should be sufficient for TEM observation. The clear observation on types in different planes of 3-D continuous double gyroid structure is more convincing, especially for the structure of extremely small 5-6 nm domain spacing.
3. On page 11, the author mentioned the poor ordering attributed to the increased number of glucose units, which reduces chain mobility. However, according to the DSC traces in Figure S11, the glass transition temperature of Glcn-b-Sol ($n = 6$ or 7) is not significantly increased, especially compared to that of Glc5-b-Sol. Instead of the 180 °C for 36 h, the author may have the bulk samples annealed at a higher temperature (for instance 220 °C), to ensure the high chain mobility in the annealing process.
4. In the SAXS profile of Glc1-b-Sol in Fig. 2, the intensity of the primary scattering signal at q^{**} marked with blue color is even lower than that of the secondary signal at $2q^{**}$. This is rarely seen and should be checked again. Besides, in Fig. 4 the high-ordered peaks of (f) and (e) attributed to GYR seem different. Especially the marked sixth peaks are not even at the same q position. Please check the assigned peaks for GYR in these two SAXS profiles again.
5. A mistake in descriptions of (b) and (c) in Fig. 2. According to the figure, it seems that (b) should be WAXS profiles, and (c) should be $qhkl$ versus $(h^2 + k^2 + l^2)^{1/2}$ plot.
6. When varying the dispersity, morphological changes were observed. Yes, the theory also does say that increased dispersity can facilitate morphologies with increasing curvatures. However, the morphology can also change as a result of changes in XN , and f . X is assumed to be constant, but did N and f remain constant, with dispersity being the only variable?

Reviewer #2 (Remarks to the Author):

This paper describes the synthesis and characterization of new class of discrete block co-oligomers derived from conjugation of hydrophobic terpene natural products with various oligosaccharides, which self-assemble (microphase separate) into ordered morphologies with sub-10 nm periodicities that may make them useful for nanopatterning and nanolithography applications. The molecules that are synthesized and characterized are clearly laid out and the studies are well-executed. Of particular note are the facts that the authors are able to access a gyroid phase in the pure amphiphiles and also in blends, where molar mass dispersity apparently plays a role in stabilizing this complex cubic network phase. This reviewer has some comments that are intended to more clearly set the stage for the work and to improve the quality of the exposition to some degree. Additionally, there are technical issues throughout the paper that must be resolved prior to further consideration for publication.

While the molecules presented in this work are new, Cushen et al. (Ref. 29) have used conceptually similar alkyne-azide [3+2]-Huisgen "click" cycloaddition reactions to conjugated similar oligosaccharides to low molecular weight polymers. Additionally, Hillmyer, Lodge, and Siepmann (Ref. 40) have demonstrated in 2019 the synthesis of diblock oligomers with that assemble into periodic cylindrical morphologies with sub-3-nm domain sizes and lamellae with sub-2 nm domains. This context is hidden in the paper in order to make the current work appear novel—this is disingenuous and frustrating to this expert in the field. The authors need to cite Ref. 29, and more importantly, Ref. 40 earlier in this paper. Also, Lawrence Sita and co-workers have published numerous recent reports on sugar-polyolefin diblocks made by catalytic chain transfer polymerization, and a few more of those references (beyond Ref. 31) should be included with comparisons of the observed domain sizes. The novelty of the current work is really in the large scale syntheses of these materials and their tunable blend phase behavior and that should be emphasized.

To emphasize, the claim on Page 16 of the smallest microphase separated LAM domain spacing of 4.2 nm is incorrect: Hillmyer, Lodge and Sipemann reported a lamellar spacing of 3.5 nm and this citation does not appear here at all.

A detailed list of comments is given below, including a series of technical issues that must be corrected prior to publication. Comments 11-15 are particularly important from a scientific standpoint and must be address without question.

1. The laundry list of polymers on Page 3 is not particularly helpful. Is there a reason that these systems are so explicitly named aside from the reference? Can these system be summarized as high chi/low diblock polymers of various chemical compositions?
2. On Page 3, it would be more useful to cite the typical dispersities and comment on how dispersity is related to the standard deviation in the mean molar mass—this calculation is outlined in Ref. 18.
3. At the top of Page 4, "Hawker et al." should really be "Bates and Hawker et al." to acknowledge the researchers who did the work.
4. The middle of Page 4 would be an appropriate place to acknowledge the work by Hillmyer, Lodge, and Siepmann as well as follow up work on the computational design of BCOs for patterning applications.
5. On Page 6, what does it mean that the spectra "were reasonably assigned to the expected BCOs"? Is there some ambiguity? The authors would do better to show the NMR of the per-acetylated product in addition to the deprotected final BCO to convince the reader that the ¹H NMR assignments are correct.

6. On Page 6, the solvent for the SEC experiments should be given in context.
7. On Page 8, how did the authors determine f_{sugar} for the calculation of the cylinder diameter? This information must be provided.
8. In Figure 2, the authors should label the "LAM" in Glc1-b-Sol as "LAM_c" to indicate crystalline lamellae to differentiate it from liquid crystalline or incompatibility-driven microphase separated lamellae.
9. In Figure 2 caption, panels (b) and (c) are interchanged—please fix.
10. Table 1 and discussion at the bottom of Page 11: "random two phase" has NO MEANING. The authors mean "melt-disordered" to indicate that there is segregation (see SAXS data) but there is no periodic long-range order.
11. The discussion on Page 11 middle regarding volume fraction is INCORRECT: lamellar $2q^*$ and $3q^*$ peaks are not entirely extinct in the SAXS patterns. Similar SAXS patterns to Glc3-b-Sol were reported by Bates and co-workers for a sample with a volume fraction of $f = 0.4$ (doi: 10.1021/ma0617421). The compositions of the current oligomers are NOT rigorously 1:2 and 1:1 as asserted due to the apparent but diminished intensities of the peaks named by the authors.
12. Page 13 bottom: the authors citation to Ref. 19 is meaningless. They should cite work by others including experimental work by Mahanthappa and co-workers (doi: 10.1021/acs.macromol.7b01452) as well as Fors (doi: 10.1038/s41578-019-0138-8), and theory by Matsen and co-workers (doi 10.1103/PhysRevLett.99.148304; doi:10.1140/epje/i2013-13044-9) and a recent review (doi: 10.1080/15583724.2019.1579227)
13. Pages 141-5: In the Glc3-b-Sol series of blends that make gyroid, the peak widths broaden with increasing dispersity. The discussion here would be much more useful if the authors conducted a simple Scherrer analysis of this SAXS data to assess how the microphase separated grain diameter decreases with increasing dispersity. This grain size is important for patterning applications.
14. Page 18 bottom–Page 19: the authors discussion of liquid crystallinity versus microphase separation is convoluted by their inclusion of molecular crystallization. In truth, the materials that they describe in this paper are liquid crystals and they do microphase separate like block polymers. There is no real dichotomy except that the authors are thinking about their system as polymer scientists and not as liquid crystal physicists.
15. Page 19: That the authors do not see the liquid crystal "clearing temperature" by DSC and the claim that this means that their systems are "polymers" is all WRONG. Order-disorder transition temperatures can be observed in diblock polymers under certain circumstances such that the DSC results are meaningless. See literature by Bates in DOI: 0.1002/aic.14023 and DOI: 0.1021/acs.macromol.5b00277

In the responses, the page and line numbers reflect those in the originally submitted version.

Reviewer #1

In this paper, a block co-oligomer (BCO) system consisting of hydrophilic sugars and hydrophobic terpenoids was synthesized by click reaction. Under thermal annealing, the resulting BCOs self-assembled into well-ordered lamellar (LAM), gyroid (GYR), and hexagonally close-packed cylindrical (HEX) nanostructures showing 4-8 nm periodicity, which are among smallest domains ever reported. These BCO nanostructures show various application potentials in nanolithography, organic devices, separation materials, coatings, and other fields.

The discovery of the self-assembled domain with a minimal 4.2 nm periodicity is appealing to the current block copolymer lithography research field. Besides, the tailoring on self-assembled morphology by changing the number of glucose or isoprene units, and tuning on Δ Glc in BCO is convincing and comprehensive. Therefore, this paper will be published in communications chemistry with minor revisions.

I have several questions as follows:

1. In Table 1, the author should explain why the annealing condition for preparation GYR morphology (100 °C, 36 h) is different from the others (130 °C, 36 h). Perhaps the GYR morphology is in a meta-stable state and a morphology transition may occur at the higher annealing temperature.

Reply to the comment: We thank the reviewer very much for this comment. We had checked the SAXS profiles for the GYR-forming Glc₂-*b*-Sol after thermal annealing at 130 °C for 36 h, and we found the scattering profile assignable to the GYR structure. However, the scattering peaks are broader compared to the one for the sample annealed at 100 °C. According to the reviewer's comment, we have added a comment on this matter at the line 4 in page 10 as follows:

“Although a similar scattering pattern assignable to the GYR phase was found in the sample after annealing at 130 °C for 36 h, the scattering peak width was broader, suggesting less ordering.”

In addition, we have added the SAXS profile for the Glc₂-*b*-Sol sample annealed at 130 °C for 36 h as **Supplementary Figure 19**.

2. The author reported a few GYR morphologies for the studied BCO. However, the morphology analysis is merely based on the detected peak ratios in the SAXS profile. To make the discovery more solid, TEM characterization is suggested, as ruthenium tetroxide may selectively stain the hydroxyl groups-rich sugar-based segments and osmium tetroxide may selectively stain the isoprene segments.

Therefore, the contrast between different domains should be sufficient for TEM observation. The clear observation on types in different planes of 3-D continuous double gyroid structure is more convincing, especially for the structure of extremely small 5-6 nm domain spacing.

Reply to the comment: We thank the reviewer very much for the insightful comment. We agree with the reviewer's comment that we should provide TEM images for the GYR-forming sample. However, at this moment, we have not yet succeeded in the TEM observation for our BCO samples due to the difficulty in the sample preparation. Such TEM characterization of the BCO samples is another big research subject. Indeed, we are currently working on direct observation (TEM and AFM) of the GYR morphology formed by Glc₂-*b*-Sol and the other relevant new materials. We would like to publish the results in future as a separate paper. Nevertheless, since the multiple scattering peaks (more than 9 peaks) observed in Fig. 2a were successfully assigned the crystallographic planes of the GYR phase (see Figure 2c), we firmly believe that Glc₂-*b*-Sol does indeed form GYR nanostructure.

3. On page 11, the author mentioned the poor ordering attributed to the increased number of glucose units, which reduces chain mobility. However, according to the DSC traces in Figure S11, the glass transition temperature of Glc_{*n*}-*b*-Sol (*n* = 6 or 7) is not significantly increased, especially compared to that of Glc₅-*b*-Sol. Instead of the 180 °C for 36 h, the author may have the bulk samples annealed at a higher temperature (for instance 220 °C), to ensure the high chain mobility in the annealing process.

Reply to the comment: We thank the reviewer very much for the intriguing comment. We agree with the reviewer's comment that we should try the annealing of Glc_{*n*}-*b*-Sol (*n* = 6 or 7) at a higher temperature. However, we did not try such experiment because it is highly expected that the prolonged heating at Temp.>180 °C leads to thermal caramelization of the oligosaccharide block (see our previous paper: *ACS Macro Lett.* **2012**, *1*, 1379). While the thermal caramelization of the oligosaccharide block may bring interesting phase behaviors depending on the annealing temperature, it makes difficult to discuss the equilibrium phase of the BCP. Therefore, we do not make any further discussion in the manuscript, in regard to the higher temperature annealing of Glc_{*n*}-*b*-Sol (*n* = 6 or 7). We hope the reviewer accepts this comment.

4. In the SAXS profile of Glc₁-*b*-Sol in Fig. 2, the intensity of the primary scattering signal at q^{**} marked with blue color is even lower than that of the secondary signal at $2q^{**}$. This is rarely seen and should be checked again. Besides, in Fig. 4 the high-ordered peaks of (f) and (e) attributed to GYR seem different. Especially the marked sixth peaks are not even at the same *q* position. Please check the assigned peaks for GYR in these two SAXS profiles again.

Reply to the comment: We thank this reviewer's comment very much. The higher intensity of $2q^{**}$

peak could be attributed to the enhancement with the intense q^* peak.

As for the peak assignment of Fig. 4 (e) and (f), we have double checked the SAXS data and firmly confirmed that those higher order peaks are indeed assignable the crystallographic planes of the GYR phase. The q_{hkl} versus $(h^2 + k^2 + l^2)^{1/2}$ plot based on the SAXS pattern validated space group assignment (see below). We have added this figure in the Supplementary Information as Supplementary Figure 21. We hope the reviewer accepts this comment.

5. A mistake in descriptions of (b) and (c) in Fig. 2. According to the figure, it seems that (b) should be WAXS profiles, and (c) should be q_{hkl} versus $(h^2 + k^2 + l^2)^{1/2}$ plot.

Reply to the comment: We thank the reviewer very much for this comment. We have corrected the figure caption.

6. When varying the dispersity, morphological changes were observed. Yes, the theory also does say that increased dispersity can facilitate morphologies with increasing curvatures. However, the morphology can also change as a result of changes in XN , and f . X is assumed to be constant, but did N and f remain constant, with dispersity being the only variable?

Reply to the comment: We thank the reviewer very much for the insightful comment. Given that the density of the sugar segment is constant, we believe that the average N and f should be constant even by changing the dispersity.

Reviewer #2

This paper describes the synthesis and characterization of new class of discrete block co-oligomers derived from conjugation of hydrophobic terpene natural products with various oligosaccharides, which self-assemble (microphase separate) into ordered morphologies with sub-10 nm periodicities that may make them useful for nanopatterning and nanolithography applications. The molecules that are synthesized and characterized are clearly laid out and the studies are well-executed. Of particular note are the facts that the authors are able to access a gyroid phase in the pure amphiphiles and also in blends, where molar mass dispersity apparently plays a role in stabilizing this complex cubic network phase. This reviewer has some comments that are intended to more clearly set the stage for the work and to improve the quality of the exposition to some degree. Additionally, there are technical issues throughout the paper that must be resolved prior to further consideration for publication.

While the molecules presented in this work are new, Cushen et al. (Ref. 29) have used conceptually similar alkyne-azide [3+2]-Huisgen “click” cycloaddition reactions to conjugated similar oligosaccharides to low molecular weight polymers. Additionally, Hillmyer, Lodge, and Siepmann (Ref. 40) have demonstrated in 2019 the synthesis of diblock oligomers with that assemble into periodic cylindrical morphologies with sub-3-nm domain sizes and lamellae with sub-2 nm domains. This context is hidden in the paper in order to make the current work appear novel—this is disingenuous and frustrating to this expert in the field. The authors need to cite Ref. 29, and more importantly, Ref. 40 earlier in this paper. Also, Lawrence Sita and co-workers have published numerous recent reports on sugar-polyolefin diblocks made by catalytic chain transfer polymerization, and a few more of those references (beyond Ref. 31) should be included with comparisons of the observed domain sizes. The novelty of the current work is really in the large scale syntheses of these materials and their tunable blend phase behavior and that should be emphasized.

Reply to the comment: We thank the reviewer very much for the helpful and valuable comments. According to the reviewer’s comment, we have revised the paragraph starting from the line 19 in page 4 as follows:

“The previous studies into oligosaccharide-based BCPs synthesized via the “click” chemistry revealed the formation of small microphase-separated structures^{26–30} with the d of even down to 5.8 nm (in the bulk state).^{31,32} Therefore, strong segregation between hydrophilic saccharidic blocks and hydrophobic isoprene-based hydrocarbon chains would be expected, which would in turn allow microphase separation in the molecular-weight regime below 2000 g mol⁻¹. In addition, recent reports^{33–35} by Lodge, Siepmann, and Hillmyer on sugar-based amphiphiles with a hydrocarbon chain, leading to a lamellar morphology with d of 3.5 nm (even down to d of 1.2 nm as revealed by molecular dynamic simulation), also supports that the combination of sugars and terpenoids is highly promising for

realizing ultrasmall nanostructure formation.”

In addition, we have revised the last two sentences of the abstract as follows:

“These BCOs microphase separated into lamellar, gyroid, and cylindrical morphologies with the domain spacing (d) of 4.2 – 7.5 nm. Considering easy synthesis and rich phase behavior, presented BCO systems could be highly promising for application to diverse ~4-nm nanofabrications.”

To emphasize, the claim on Page 16 of the smallest microphase separated LAM domain spacing of 4.2 nm is incorrect: Hillmyer, Lodge and Sipemann reported a lamellar spacing of 3.5 nm and this citation does not appear here at all.

Reply to the comment: According to the reviewer’s comment, we have revised the sentence at the line 10 in page 16 as follows together with the suggested citation:

“Such ultrasmall d value of ~4 nm has rarely been reported among the organic-BCP-derived LAM microphase-separated structures.^{9,14,23,31,34,47}”

A detailed list of comments is given below, including a series of technical issues that must be corrected prior to publication. Comments 11-15 are particularly important from a scientific standpoint and must be address without question.

1. The laundry list of polymers on Page 3 is not particularly helpful. Is there a reason that these systems are so explicitly named aside from the reference? Can these system be summarized as high χ /low diblock polymers of various chemical compositions?

Reply to the comment: We thank the reviewer very much for this comment. According to the reviewer’s comment, we have removed the sentence at the line 12 in page 3 and those citations were included in the previous sentence.

2. On Page 3, it would be more useful to cite the typical dispersities and comment on how dispersity is related to the standard deviation in the mean molar mass–this calculation is outlined in Ref. 18.

Reply to the comment: We thank the reviewer very much for this comment. According to the reviewer’s comment, we have revised the sentence at the line 18 in page 3 as follows:

“As a result, it is virtually impossible for all growing polymer chains to terminate at the same length;

therefore, the BCPs exhibit a range of molecular weight distributions (typically, dispersity of 1.05–1.30).⁹⁻¹⁶”

3. At the top of Page 4, “Hawker et al.” should really be “Bates and Hawker et al.” to acknowledge the researchers who did the work.

Reply to the comment: We thank the reviewer very much for this comment. According to the reviewer’s comment, we have revised the sentence at the line 2 in page 4 as follows:

“Indeed, seminal works by Meijer *et al.*²¹ and Bates and Hawker *et al.*²² have demonstrated that d increased with increasing BCP dispersity.”

4. The middle of Page 4 would be an appropriate place to acknowledge the work by Hillmyer, Lodge, and Siepman as well as follow up work on the computational design of BCOs for patterning applications.

Reply to the comment: We thank the reviewer very much for this comment. According to the reviewer’s comment, we have newly added the following sentence at the bottom of page 4:

“In addition, recent reports³³⁻³⁵ by Lodge, Siepman, and Hillmyer on sugar-based amphiphiles with a hydrocarbon chain, leading to a lamellar morphology with d of 3.5 nm (even down to d of 1.2 nm as revealed by molecular dynamic simulation), also supports that the combination of sugars and terpenoids is highly promising for realizing ultrasmall nanostructure formation.”

5. On Page 6, what does it mean that the spectra “were reasonably assigned to the expected BCOs”? Is there some ambiguity? The authors would do better to show the NMR of the per-acetylated product in addition to the deprotected final BCO to convince the reader that the ¹H NMR assignments are correct.

Reply to the comment: The NMR spectra of the BCOs are all assignable to the expected chemical structures, as shown in Supplementary Figures 2-8. We have removed the “reasonably” from the sentence at the line 6 in page 6 to avoid misleading. We firmly believe that the SEC, IR, ¹H NMR, and MALDI-TOF MS analyses have already provided concrete evidences that the obtained materials are indeed the desired compounds.

6. On Page 6, the solvent for the SEC experiments should be given in context.

Reply to the comment: We thank the reviewer very much for this comment. According to the reviewer's comment, we have revised the sentence at the line 5 in page 6 as follows:

“Proton nuclear magnetic resonance (^1H NMR) spectra were assigned to the expected BCO chemical structure (Supplementary Figures 2–8), and the size-exclusion-chromatography (SEC; in DMF containing 0.01 M LiCl) elution peak clearly shifted toward the higher-molecular-weight region upon increasing the number of glucose units in the BCO (Fig. 1a).”

In addition, we have added the SEC solvent information in the figure caption of Fig. 1.

7. On Page 8, how did the authors determine f_{sugar} for the calculation of the cylinder diameter? This information must be provided.

Reply to the comment: We thank the reviewer very much for this question. According to the reviewer's comment, we have revised the sentence at the line 7 in page 8 as follows:

“It is rather surprising that even the BCO containing only one glucose unit, *i.e.*, Glc₁-*b*-Sol, had self-assembled into the HEX morphology with a domain spacing ($d = 2\pi/q^*$) of 5.5 nm (cylinder-to-cylinder distance = $2d/\sqrt{3} = 6.4$ nm; cylinder diameter = $[8f_{\text{sugar}}d^2/(\sqrt{3}\cdot\pi)]^{1/2} = 2.7$ nm, where f_{sugar} is the saccharide volume fraction calculated based on the densities of 1.36 and 0.95 g cm⁻³ for the saccharide and solanesol blocks, respectively),²³ which is among the smallest ever obtained among structures that have been self-assembled from saccharide-based block BCPs.”

8. In Figure 2, the authors should label the “LAM” in Glc1-*b*-Sol as “LAM_c” to indicate crystalline lamellae to differentiate it from liquid crystalline or incompatibility-driven microphase separated lamellae.

Reply to the comment: We thank the reviewer very much for this comment. According to the reviewer's comment, we have added a new sentence at the line 18 in page 8 as follows:

“We thus denote this nanostructure as LAM_c to distinguish it with the microphase-separated LAM structure.”

In addition, we have replaced the expressions of “LAM” in Table 1 and Figure 1a by “LAM_c”.

9. In Figure 2 caption, panels (b) and (c) are interchanged—please fix.

Reply to the comment: We thank the reviewer very much for this comment. According to the reviewer’s comment, we have revised the caption of Figure 2.

10. Table 1 and discussion at the bottom of Page 11: “random two phase” has NO MEANING. The authors mean “melt-disordered” to indicate that there is segregation (see SAXS data) but there is no periodic long-range order.

Reply to the comment: We thank the reviewer very much for this comment. According to the reviewer’s comment, we have replaced the “random two phase” in Table 1 by “melt-disordered”. In addition, the expressions of the “Random two phase” in Fig. 2 were also replaced by “melt-disordered”.

11. The discussion on Page 11 middle regarding volume fraction is INCORRECT: lamellar $2q^*$ and $3q^*$ peaks are not entirely extinct in the SAXS patterns. Similar SAXS patterns to Glc3-b-Sol were reported by Bates and co-workers for a sample with a volume fraction of $f = 0.4$ (doi: 10.1021/ma0617421). The compositions of the current oligomers are NOT rigorously 1:2 and 1:1 as asserted due to the apparent but diminished intensities of the peaks named by the authors.

Reply to the comment: We thank the reviewer very much for this insightful comment. With having this comment, we have removed the sentence discussing the volume fraction based on the SAXS results.

12. Page 13 bottom: the authors citation to Ref. 19 is meaningless. They should cite work by others including experimental work by Mahanthappa and co-workers (doi: 10.1021/acs.macromol.7b01452) as well as Fors (doi: 10.1038/s41578-019-0138-8), and theory by Matsen and co-workers (doi 10.1103/PhysRevLett.99.148304; doi:10.1140/epje/i2013-13044-9) and a recent review (doi: 10.1080/15583724.2019.1579227)

Reply to the comment: We thank the reviewer very much for the helpful comments. We have removed the Ref. 19 from the sentence at the line 3 in page 13 and have cited the suggested literatures.

13. Pages 141-5: In the Glc3-b-Sol series of blends that make gyroid, the peak widths broaden with increasing dispersity. The discussion here would be much more useful if the authors conducted a simple Scherrer analysis of this SAXS data to assess how the microphase separated grain diameter decreases with increasing dispersity. This grain size is important for patterning applications.

Reply to the comment: We thank the reviewer very much for the helpful and valuable comments. With having the reviewer's comment, we analyzed the full width at half maximum (FWHM, Δq^*) for the primary scattering peak of the GYR-forming samples. As a result, we found a general tendency that the FWHM, which is inversely proportional to the grain size according to the Scherrer's equation, increased with increasing D_{Glc} (e.g., $\Delta q^* = 0.0326 \text{ nm}^{-1}$ for $D_{\text{Glc}} = 1.069$; $\Delta q^* = 0.0365 \text{ nm}^{-1}$ for $D_{\text{Glc}} = 1.244$). This indicated that the grain size of the GYR morphology decreased with the increasing D_{Glc} . To address this issue, we have newly added the following sentences at the line 10 in page 14:

“Interestingly, for the GYR-forming samples, the full width at half maximum (Δq^*) of the primary scattering peak increased with increasing D_{Glc} (e.g., $\Delta q^* = 0.0326 \text{ nm}^{-1}$ for $D_{\text{Glc}} = 1.069$; $\Delta q^* = 0.0365 \text{ nm}^{-1}$ for $D_{\text{Glc}} = 1.244$). Since the Δq^* is inversely proportional to the grain size according to the Scherrer's equation,⁴⁶ this tendency implies the decrease in the GYR grain size with increasing D_{Glc} .”

14. Page 18 bottom–Page 19: the authors discussion of liquid crystallinity versus microphase separation is convoluted by their inclusion of molecular crystallization. In truth, the materials that they describe in this paper are liquid crystals and they do microphase separate like block polymers. There is no real dichotomy except that the authors are thinking about their system as polymer scientists and not as liquid crystal physicists.

15. Page 19: That the authors do not see the liquid crystal “clearing temperature” by DSC and the claim that this means that their systems are “polymers” is all WRONG. Order-disorder transition temperatures can be observed in diblock polymers under certain circumstances such that the DSC results are meaningless. See literature by Bates in DOI: 0.1002/aic.14023 and DOI: 0.1021/acs.macromol.5b00277

Reply to the comment: We thank the reviewer very much for those insightful comments. With having the reviewer's comments, we decided to remove the last paragraph of the Results and Discussion section.

Nonetheless, it would still be useful to discuss the DSC results of the BCO system. Therefore, we added the following sentences at the line 4 in page 8 to address the DSC results.

“Prior to the SAXS experiment, we performed differential scanning calorimetry (DSC) analysis on the BCOs. Although all the BCO DSC thermograms did not exhibit any transitions due to crystallization and melting during the first cooling and second heating processes, they did exhibit two baseline shifts corresponding to glass transition temperatures (T_{gs}) for the solanesol and sugar segments (**Supplementary Figure 16**). The fact that the T_{gs} of the two segments were independently observed suggested the microphase separation between the segments.”

Additional Changes:

1. We have revised the sentence at the line 18 in page 3 as follows:

“However, the high χ -low N BCPs are synthesized partially or entirely by polymerization reactions.”

2. We have added a new sentence at the line 7 in page 11 as follows:

“It should be notable that Glc₃-*b*-Sol formed the LAM despite the asymmetric composition ($f_{\text{sugar}} = 0.33$).”

3. We found that we cited the *Poly. Chem.* **2019**, *10*, 1119 twice in the main text (Ref. 16 and 36). We have cited another suitable literature as Ref. 16.

Ref. 16: Katsuhara, S.; Mamiya, H.; Yamamoto, T.; Tajima, K.; Isono, T.; Satoh, T. Metallopolymer-*block*-oligosaccharide for sub-10 nm microphase separation. *Polym. Chem.* **2020**, *11*, 2995-3002.

Reviewers' comments:

Reviewer #1 (Remarks to the Author):

The manuscript by T. Satoh et al. reports the Combining Sugar and Terpenoid: Rapid Access to Discrete and Monodisperse Block Co-oligomers for Ultrasmall Periodic Nanostructures. The authors have prepared the revised manuscript according to the reviewer's comments with a point-to-point reply. The revision is now suitable for publication in Communications Chemistry.

Reviewer #3 (Remarks to the Author):

I have checked the author's response letter, and found that they already gave the clear and careful replies to reviewer's comments. Few reversions are still needed to publish this manuscript in communications chemistry.

1. Page 11, "according to classical BCP theory, 1/2 volume fractions should lead to HEX rather than LAM structural formations ". here the 1/2 should be 1/3, at least for most of the block copolymers.
2. In comparison with thermal annealing, solvent vapor annealing (SVA) is more suitable for the polymers which degrade/ cross-link or burn easily. In addition, the SVA will endow Glc6-b-Sol and Glc7-b-Sol with sufficient mobility to form ordered structures considered that they show a larger segregation strength (χN). The author should explain why thermal annealing is used instead of the SVA for those sugar contained block co-oligomers.
3. It is better to provide the equilibrium nanostructure of Glc6-b-Sol and Glc7-b-Sol to indicate that the continuous evolution of self-assembled structures are directed by the changes of compositions and molecular weights.

In the responses, the page and line numbers reflect those in the originally submitted version.

Reviewer #1

The manuscript by T. Satoh et al. reports the Combining Sugar and Terpenoid: Rapid Access to Discrete and Monodisperse Block Co-oligomers for Ultrasmall Periodic Nanostructures. The authors have prepared the revised manuscript according to the reviewer's comments with a point-to-point reply. The revision is now suitable for publication in Communications Chemistry.

Reply to the comment: We thank the reviewer very much for carefully reading our manuscript again.

Reviewer #3

I have checked the author's response letter, and found that they already gave the clear and careful replies to reviewer's comments. Few reversions are still needed to publish this manuscript in communications chemistry.

1. Page 11, "according to classical BCP theory, 1/2 volume fractions should lead to HEX rather than LAM structural formations " here the 1/2 should be 1/3, at least for most of the block copolymers.

Reply to the comment: We thank the reviewer very much for this comment. With having this comment, we have revised the sentence as follows:

“According to classical BCP theory, **volume fraction of ~0.3** should lead to HEX rather than LAM structural formations.”

2. In comparison with thermal annealing, solvent vapor annealing (SVA) is more suitable for the polymers which degrade/ cross-link or burn easily. In addition, the SVA will endow Glc6-b-Sol and Glc7-b-Sol with sufficient mobility to form ordered structures considered that they show a larger segregation strength (χN). The author should explain why thermal annealing is used instead of the SVA for those sugar contained block co-oligomers.

Reply to the comment: We thank the reviewer very much for the insightful comment. Actually, we had reached the same idea as suggested by the reviewer. In fact, we had tried to the solvent annealing using *N,N*-dimethylformamide vapor for the self-assembly of those BCOs in the bulk state. However,

no ordered nanostructure was found in the solvent-annealed samples. Therefore, we do not make any further discussion in the manuscript in regard to the solvent annealing. We hope the reviewer accepts this.

3. It is better to provide the equilibrium nanostructure of Glc6-*b*-Sol and Glc7-*b*-Sol to indicate that the continuous evolution of self-assembled structures are directed by the changes of compositions and molecular weights.

Reply to the comment: We thank the reviewer very much for this comment. As mentioned above, we have so far failed to find a suitable annealing condition to produce the equilibrium nanostructures from Glc₆-*b*-Sol and Glc₇-*b*-Sol. To discuss about the equilibrium nanostructures of Glc₆-*b*-Sol and Glc₇-*b*-Sol, further screening of the annealing condition is needed. Since those two BCOs have higher saccharide volume fraction, we expect that they could self-assemble into the microphase-separated structures with the saccharide matrix and hydrocarbon microdomain. Such morphology has rarely been investigated, which requires detailed further investigations. We hope to publish results in near future in a separate paper.

Additional Changes:

1. We have revised the sentence at the line 13 in page 8 as follows:

“It is rather surprising that even the BCO containing only one glucose unit, *i.e.*, Glc₁-*b*-Sol, had self-assembled into the HEX morphology with a domain spacing ($d = 2\pi/q^*$) of 5.5 nm (cylinder-to-cylinder distance = $2d/\sqrt{3} = 6.4$ nm; cylinder diameter = $[8f_{\text{sugar}} \cdot d^2/(\sqrt{3} \cdot \pi)]^{1/2} = 2.7$ nm, where f_{sugar} is the **saccharide** volume fraction calculated based on the densities of 1.36 and 0.95 for the saccharide and solanesol blocks, respectively),²³ which is among the smallest ever obtained among structures that have been self-assembled from saccharide-based block BCPs”

2. We have revised the first sentence in the Acknowledgement section as follows:

“This work was financially supported by a JSPS Grant-in-Aid for Scientific Research (B) (No. 19H02769, T. S.; No. 20H02792, T. I.), a JSPS Grant-in-Aid for Young Scientists (No. 18K14268, T. I.), the Photoexcitonix Project (Hokkaido University, T. S.), the Frontier Chemistry Center (Hokkaido University, T. S. and T. I.), **the Creative Research Institute (CRIS; Hokkaido University, T. S.)**, the Tonen General Sekiyu Research/Development Encouragement & Scholarship Foundation (T. I.), the Shorai Foundation for Science and Technology (T. I.), the Iketani Science and Technology Foundation (T. I.), and the Asahi Glass Foundation (T. I.).”